# A marine photosynthetic microbial cell factory as a platform for spider silk production

Choon Pin Foong[1,2], Mieko Higuchi-Takeuchi[1], Ali D. Malay [1], Nur Alia Oktaviani[1], Chonprakun Thagun[1] & Keiji Numata [1,2✉]

Photosynthetic microorganisms such as cyanobacteria, purple bacteria and microalgae have attracted great interest as promising platforms for economical and sustainable production of bioenergy, biochemicals, and biopolymers. Here, we demonstrate heterotrophic production of spider dragline silk proteins, major ampullate spidroins (MaSp), in a marine photosynthetic purple bacterium, *Rhodovulum sulfidophilum*, under both photoheterotrophic and photo-autotrophic growth conditions. Spider silk is a biodegradable and biocompatible material with remarkable mechanical properties. *R. sulfidophilum* grow by utilizing abundant and renewable nonfood bioresources such as seawater, sunlight, and gaseous $CO_2$ and $N_2$, thus making this photosynthetic microbial cell factory a promising green and sustainable production platform for proteins and biopolymers, including spider silks.

[1] Biomacromolecules Research Team, RIKEN Center for Sustainable Resource Science, 2-1 Hirosawa, Wako, Saitama 351-0198, Japan. [2] Department of Material Chemistry, Graduate School of Engineering, Kyoto University, Kyoto-Daigaku-Katsura, Nishikyo-ku, Kyoto 615-8510, Japan. ✉email: keiji.numata@riken.jp

Growing awareness of climate change, depletion of non-renewable fossil resources, and global food and water crises have recently spurred efforts to develop "sustainable cell factory" platforms for the production of valuable biocompounds/chemicals. Ideally, these next-generation cell factories should employ eco-friendly and sustainable bioprocesses and solely depend on renewable nonfood bioresources as feedstocks. We have been developing a purple nonsulfur bacterium, *Rhodovulum sulfidophilum*, that confers advantages from both photosynthetic[1,2], and halophilic[3,4] abilities as a potential alternative workhorse to replace current heterotrophic microbial cell factories[5]. *R. sulfidophilum* is a marine anoxygenic photosynthetic bacterium with versatile metabolic capabilities that produces biohydrogen[6], bioplastic[7], and extracellular nucleic acids[8]. The most important points are its ability to grow under photoautotrophic conditions by utilizing low-cost and abundant renewable resources such as light (energy), $CO_2$ (carbon source), and $N_2$ (nitrogen source) via photosynthesis and nitrogen fixation processes[9–11] and its ability to grow in seawater, which could lower the risk of biological contamination during cultivation[5].

Nature provides extremely strong and tough biomaterials, such as spider silk[12], limpet teeth[13], and bagworm silk[14]. Spider dragline silk, in particular, has been extensively studied due to its outstanding features, including high tensile strength, high extensibility, and low weight[15,16]. In addition, the biodegradable and biocompatible features of spider dragline silk have made it suitable for biomedical and eco-friendly applications[17]. Major ampullate spidroin (MaSp) is produced in the major ampullate gland of spiders, and spun silk fibers are mainly composed of multiple types of MaSp, such as MaSp1 and MaSp2[18,19]. MaSp has a conserved primary structure comprising three domains: a repetitive central domain and nonrepetitive *N*-terminal and *C*-terminal domains. The MaSp repetitive domains are arranged in alternating blocks of polyalanine (crystalline) and glycine-rich (amorphous) sequences, which are responsible for the high tensile strength and high elasticity, respectively, of spider silk fibers[20,21].

Current mass production of spidroins has been achieved using recombinant host organisms because of low yields from spider silk glands and the cannibalistic and territorial nature of spiders[22,23]. Spidroins have been successfully expressed in recombinant bacteria (*Escherichia coli*)[24,25], yeasts (*Pichia pastoris*)[26], insects (silkworm *Bombyx mori*)[27], plants (tobacco and potato)[28], and animals (mice and mammalian cell cultures)[29,30]. Using bacterial or yeast fermentation technologies, a few venture companies have launched various prototypes made of artificial spider silk fibers[31]. However, it is still a great challenge to produce spidroins on a large scale with a sustainable production process, even though spider silk is an eco-friendly and sustainable material. Moreover, the hydrophobic tandem sequences of MaSp1 could reduce productivity by microbial fermentation. Besides, high price of spider silk due to high production cost also remains a challenge to be resolved. Raw materials that used in heterotrophic microbial fermentation systems could contribute up to 70% of production cost[32].

Here, we develop an economical and sustainable marine photosynthetic microbial cell factory using *R. sulfidophilum*, which is a marine purple nonsulfur bacterium that is capable of producing the hydrophobic repetitive sequence of MaSp1 using small amount of organic substance under photoheterotrophic or photoautotrophic growth conditions. Although very little information is available for recombinant protein expression in *R. sulfidophilum* except for studies related to its photosynthetic apparatus[33,34]. To the best of our knowledge, this is the first report of heterologous spidroin production using photosynthetic and halophilic bacteria with abundant carbon and nitrogen sources under seawater conditions.

## Results and discussion

**Construction of MaSp1-expressing *R. sulfidophilum*.** The introduction of exogenous plasmid DNA into *R. sulfidophilum* via bacterial conjugation using pCF1010-derived plasmids and *E. coli* S17-1 as a donor strain was reported[34]. This transformation was achieved based on the RP4/RK2 mating system. In this study, we used another broad-host-range vector, pBBR1MCS-2, harboring a kanamycin resistance gene, *mob* (mobility) gene and transfer origin (*oriT*), which have been widely used in Gram-negative bacterial conjugation[35,36]. In the chromosome of *R. sulfidophilum* (accession no. NZ_CP015418), two tellurite resistance genes encoding the TerB-family tellurite resistance protein were present at the loci 'A6W98_RS06280' and 'A6W98_RS17070'. Both kanamycin and tellurite resistance features were used as selection markers to distinguish positive conjugants of *R. sulfidophilum*. The newly constructed pBBR1-P$_{trc}$-MaSp1 plasmid contained (i) a *trc* promoter (P$_{trc}$), which is a hybrid (*trp* and *lacUV5* promoters, differs from *tac* promoter by 1 bp) constitutive strong promoter in *E. coli*[37], (ii) the ribosome-binding site (RBS) sequence "AGGAGA", which is derived from the upstream region of the *puf* operon (encoding a light-harvesting protein and a reaction center complex) in *R. sulfidophilum*[38], and (iii) a repetitive domain sequence of the *MaSp1* gene from *Nephila clavipes*, which had been codon-optimized for *E. coli*[24] (Fig. 1a, b, Supplementary Table 2). This gene cassette was located in the multiple cloning site of pBBR1MCS-2 but in the opposite direction of the *lac* promoter (P$_{lac}$) to avoid the influence of the *lac* promoter on our target protein expression.

**Photoheterotrophic production of different sizes MaSp1.** Approximately 0.4 g of cell wet mass (CWM) was obtained from 50 mL of a recombinant *R. sulfidophilum* culture grown to the stationary growth phase under photoheterotrophic conditions, namely, marine broth (MB) with LED illumination at 730 nm and irradiation at 20–30 W m$^{-2}$, for 4 days. Although the overexpression of the recombinant MaSp1 proteins was not detected clearly in all the recombinant *R. sulfidophilum* cultures by SDS-PAGE (Fig. 1c), we confirmed the positive expression of the MaSp1 proteins for all the newly constructed recombinant *R. sulfidophilum* cells harboring pBBR1-P$_{trc}$-MaSp1-(1-mer, 2-mer, 3-mer, or 6-mer) by western blotting (Fig. 1d) and liquid chromatography–tandem mass spectrometry (LC–MS/MS) analyses (Supplementary Data 1[39]). The single repetitive domain in our constructs contains 33 amino acid residues as follows: NH$_2$-SGRGGLGGQGAGAAAAAGGAGQGGYGGLGSQGT-COOH. The theoretical molecular weights for the target proteins, including nonspidroin sequences (His-Tag, S-Tag, enterokinase, and thrombin cleavage sites) at the *N*-terminus, are 7.9 kDa for the 1-mer (81 aa), 10.5 kDa for the 2-mer (114 aa), 13.1 kDa for the 3-mer (147 aa), and 20.9 kDa for the 6-mer (246 aa). Indeed, all the target protein bands in western blots appeared at slightly higher positions than their corresponding theoretical molecular weights. This gel shifting is due to the hydrophobicity of silk proteins in general, which affects protein–SDS interactions to reduce gel mobility[40,41]. In addition to the confirmation of MaSp1 proteins expression, we also performed a brief estimation of the amount of MaSp1 proteins obtained from the recombinant *R. sulfidophilum* cultures, which was ~3–10 mg L$^{-1}$ (1-mer = 3.4 mg L$^{-1}$, 2-mer = 3.9 mg L$^{-1}$, 3-mer = 10.2 mg L$^{-1}$, and 6-mer = 6.8 mg L$^{-1}$) or 3.5–6.9% of total proteins based on western blotting semiquantification (Supplementary Fig. 1). For comparison, heterologous expression of spidroins in a well-established and widely used recombinant *E. coli* system was able to produce ~0.3–1.2 g L$^{-1}$ purified spidroin[23,42]. Nevertheless, to our knowledge, this was the first report of successful biosynthesis of

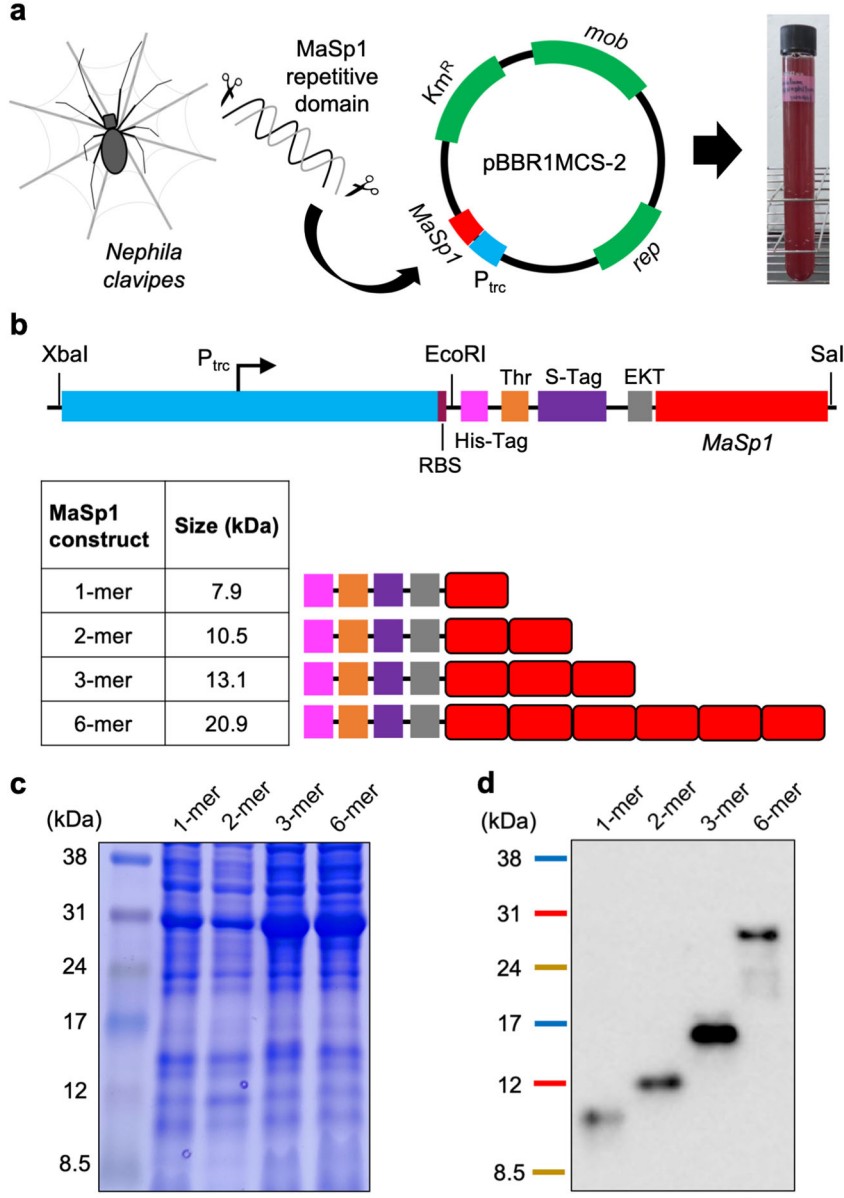

**Fig. 1 Heterologous expression of spider dragline silk proteins in the recombinant marine photosynthetic bacterium *Rhodovulum sulfidophilum* under photoheterotrophic conditions. a** A recombinant *R. sulfidophilum* harboring the broad-host-range vector pBBR1MCS-2 with a MaSp1 repetitive domain from *Nephila clavipes* was developed to express spider dragline silk protein. **b** A gene cassette containing the *trc* promoter (P$_{trc}$) and *MaSp1*-(1-mer, 2-mer, 3-mer, and 6-mer) was inserted into pBBR1MCS-2, and a histidine tag was present at the *N*-terminus of MaSp1 (pink-color box). **c** Tris-Tricine SDS-PAGE (16.5%) of soluble proteins from four days of recombinant *R. sulfidophilum* cultures. **d** Western blot using monoclonal anti-His•Tag antibody, which targets histidine-tagged MaSp1-(1-mer, 2-mer, 3-mer, or 6-mer) proteins.

artificial spider silk proteins in a marine photosynthetic bacterium under photoheterotrophic conditions. Further attempts in expressing artificial spider silk proteins with sizes close to the native spider dragline silk (~100-mer or ~300 kDa), which had been achieved in metabolically engineered *E. coli* would be applicable in *R. sulfidophilum* as well. However, there are many challenges need to be resolved for the host in advance such as metabolic capability (high demand for glycine and alanine tRNAs) and stability of the genetic constructs (long and highly repetitive DNA sequences)[42,43].

**Photoautotrophic growth and heterotrophic MaSp1 production.** The most remarkable result of this study is the demonstration of next-generation microbial cell factories based on marine photosynthetic organisms in which we can apply an

photoautotrophic growth mode by using renewable nonfood feedstocks and seawater as the cultivation medium. *R. sulfidophilum* harboring pBBR1-P$_{trc}$-MaSp1-(6-mer) was cultured in Daigo's artificial seawater (ASW) medium with light from LEDs (730 nm, 20–30 W m$^{-2}$) with a bicarbonate salt (1 g L$^{-1}$) as an inorganic carbon source and nitrogen gas (0.5 L d$^{-1}$) as a nitrogen source for 7 days (Fig. 2a). The largest repeat, MaSp1-(6-mer), was chosen for subsequent experiments because higher molecular weight of MaSp1 would contribute more tensile strength to the spider silk fiber. Sodium bicarbonate was used to supply inorganic carbon because bicarbonate salts have greater solubility and lower logistic and transportation costs than gaseous $CO_2$[44].

In our previous study, we had examined the cell growth of *R. sulfidophilum* under different light conditions, such as intensity

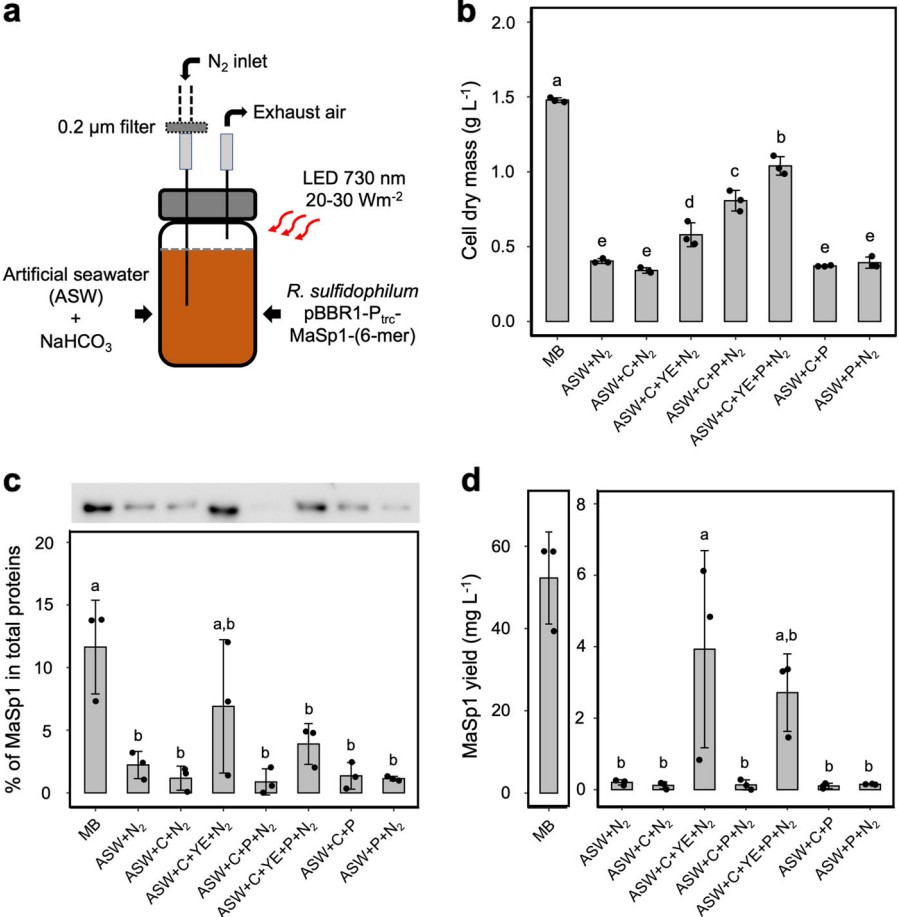

**Fig. 2 Photoautotrophic growth and heterotrophic production of artificial spider silk protein in the recombinant marine photosynthetic bacterium**
***Rhodovulum sulfidophilum.*** **a** Recombinant *R. sulfidophilum* harboring pBBR1-$P_{trc}$-MaSp1-(6-mer) was cultivated using 20 mL of Daigo's artificial seawater (ASW) SP for marine microalgae medium in a 20 mL glass vial with a rubber stopper at 30 °C with continuous far-red LED light (730 nm, 20–30 W m$^{-2}$) for 7 days. Inorganic carbon was supplied as 1 g L$^{-1}$ sodium bicarbonate, while nitrogen was supplied via daily nitrogen gas bubbling at 0.5 L d$^{-1}$. Both marine broth (MB) and ASW media contained 100 mg L$^{-1}$ kanamycin. **b** Biomass accumulation of recombinant *R. sulfidophilum* in various medium compositions based on cell dry mass (CDM). **c** Semiquantitative quantification of MaSp1-(6-mer) expression in crude cell lysate of *R. sulfidophilum* by western blot using a monoclonal anti-His●Tag antibody. **d** MaSp1-(6-mer) yield by recombinant *R. sulfidophilum* in various medium compositions. Mean data (±SD) accompanied by different letters are significantly different with *p* values < 0.05 (*n* = 3 independent biological replicates). (C = NaHCO$_3$, YE = 0.4 g L$^{-1}$ yeast extract, N$_2$ = nitrogen gas, and P = 0.5 g L$^{-1}$ KH$_2$PO$_4$).

(8 and 50 W m$^{-2}$) and wavelength (730, 800, and 850 nm)[45]. In this study, we evaluated the effect of a few additional nutrients (yeast extract, vitamin, iron, and phosphorus) that are deficient in ASW medium on the growth of recombinant *R. sulfidophilum*. The cell dry mass (CDM) decreased from 0.90 g L$^{-1}$ (with all nutrients) to 0.66 g L$^{-1}$ and 0.39 g L$^{-1}$ in the absence of yeast extract and phosphorus, respectively (Supplementary Fig. 2). In subsequent experiments, we also observed that the recombinant *R. sulfidophilum* was unable to grow in ASW medium without the supply of any of NaHCO$_3$, N$_2$ gas, or phosphorus (Fig. 2b, ASW + N$_2$, ASW + C + N$_2$, ASW + C + P, and ASW + P + N$_2$). The CDM (~0.4 g L$^{-1}$) in these ASW cultures was most likely from the inoculums or seed cultures (MB) even after the samples were washed with 2% sodium chloride. Thus, carbon, nitrogen, and phosphorus sources are all necessary for the growth of recombinant *R. sulfidophilum* in ASW medium. As expected, the cell growth increased significantly from 0.34 ± 0.02 g L$^{-1}$ (ASW + C + N$_2$) to 0.58 ± 0.08 g L$^{-1}$ (1.7-fold increase) and 0.81 ± 0.02 g L$^{-1}$ (2.4-fold increase) in the presence of yeast extract (ASW + C + N$_2$ + YE) and phosphorus (ASW + C + N$_2$ + P), respectively. The highest CDM was achieved by adding together yeast extract and phosphorus, which yielded 1.04 ± 0.06

g L$^{-1}$ (3.1-fold increase) or almost 70% of the CDM in nutrient-rich MB medium (1.48 ± 0.01 g L$^{-1}$).

An ~0.2 mg L$^{-1}$ recombinant MaSp1 protein yield and an MaSp1 content accounting for 2% of total proteins were observed in ASW + N$_2$, ASW + C + N$_2$, ASW + C + P, and ASW + P + N$_2$ (Fig. 2c, d and Supplementary Fig. 3), which might be carry-over from inoculum as explained in the previous section. MaSp1 protein production was promoted by the addition of yeast extract, which significantly increased the yield of MaSp1 protein from 0.12 ± 0.10 mg L$^{-1}$ (ASW + C + N$_2$) to 3.93 ± 2.76 mg L$^{-1}$ (ASW + C + YE + N$_2$). Yeast addition also increased the percentage of MaSp1 in the total protein from 1.2 ± 1.0 to 6.9 ± 5.3%. Interestingly, the addition of phosphorus had an adverse effect on MaSp1 protein production even though it could significantly promote CDM increments. Compared to growth in ASW + C + YE + N$_2$, growth in ASW + C + YE + P + N$_2$ decreased the yield of MaSp1 protein to 2.71 ± 1.09 mg L$^{-1}$ and the percentage of MaSp1 in total proteins to 3.9 ± 1.6%. These results could be explained by the function of each component, where the yeast extract (autolyzed yeast cells) is mainly a nitrogen source, which promotes protein biosynthesis[46–48]. Meanwhile, phosphorus is an essential macronutrient and heteroelement in many important

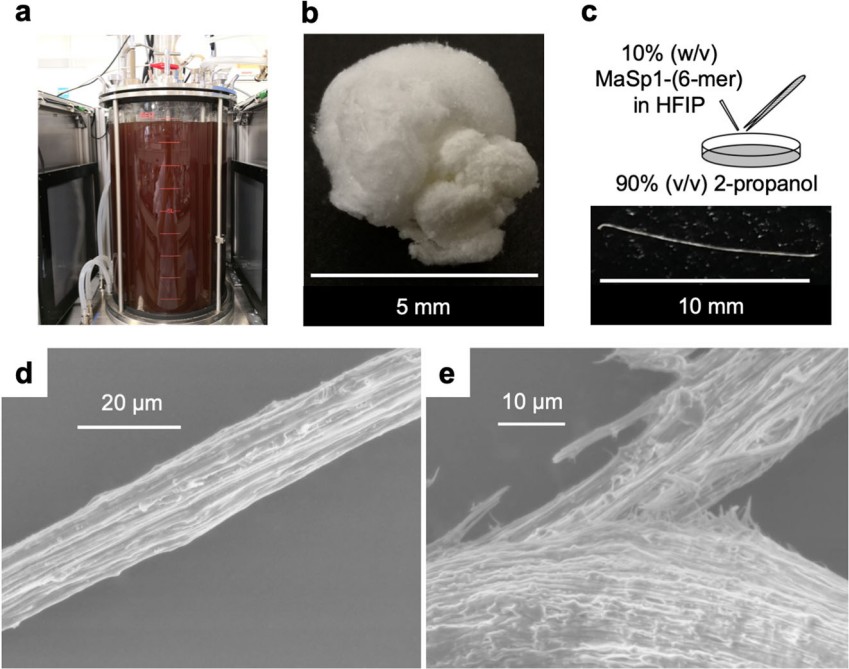

**Fig. 3 Large-scale production and fiber extrusion of MaSp1-(6-mer) artificial spidroin. a** Nine-liter-scale production of MaSp1-(6-mer) using marine broth containing 100 mg L$^{-1}$ kanamycin under photoheterotrophic conditions and continuous far-red LED light (850 nm, 15 W m$^{-2}$) at 30 °C for 7 days. **b** Lyophilization of pure MaSp1-(6-mer) after His-Tag affinity and gel filtration chromatographic purifications. **c** Fiber extrusion was performed via hand-drawing using forceps with 10% (w/v) purified MaSp1-(6-mer) dissolved in HFIP, while 2-propanol was used as a coagulation bath. **d** Scanning electron micrographs of the hand-drawn spider silk fibers at the surface. **e** Scanning electron micrographs of the break point of the spider silk fiber.

cellular compounds that promotes the growth of primary producers[49,50]. Although further optimization on the ASW medium is necessary to achieve cell growth and MaSp1 yield comparable to those in MB medium (CDM = 1.48 ± 0.01 g L$^{-1}$; MaSp1 yield = 52.28 ± 11.20 mg L$^{-1}$), we demonstrated photoautotrophic growth and heterotrophic synthesis of silk proteins by using mainly renewable nonfood feedstocks, small amount of organic substance, and seawater as the cultivation medium.

**Purification of MaSp1 for spider silk fiber formation**. To obtain a sufficient amount of MaSp1 protein for fiber extrusion, we performed nine-liter-scale jar fermentation for the production of MaSp1-(6-mer), the largest repeat available in this study (Fig. 3a). In general, the size of the spidroins have positive correlation to tensile strength until a certain size of molecular weight. Large proteins possess more interchain and intrachain interactions, more entanglements, and less chain-end defects[42,43,51]. Purifications of MaSp1-(6-mer) were carried out using affinity chromatography via histidine tag, which was present at the *N*-terminus of the *MaSp1* gene cassette (Supplementary Fig. 4a), and gel filtration chromatography (Supplementary Fig. 4b). The purified MaSp1-(6-mer) appeared in eluent fractions 1 and 2 after His-Tag purification. Purified MaSp1-(6-mer) in the eluent fractions 10–12 was combined after gel filtration and then subjected to desalting and lyophilization. In the end, we obtained ~10 mg of purified MaSp1-(6-mer) (Fig. 3b) from ~40 g of CWM. Silk fibers were produced by pipetting 10 wt% purified MaSp1-(6-mer) dissolved in hexafluoroisopropanol (HFIP) into a coagulation bath, followed by hand-drawing using forceps (Fig. 3c). The best results were obtained using 90% (v/v) 2-propanol as the coagulation bath, which induced relatively mild dehydration that allowed efficient drawing[52]. Analysis using scanning electron microscopy showed that the fibers exhibit approximately constant diameters of 10–20 μm and a surface marked by striations parallel

to the fiber axis. Fracture surface analysis revealed an internal structure consisting of microfibrils (Fig. 3d, e).

In conclusion, we have successfully established a promising marine photosynthetic microbial cell factory using the purple nonsulfur bacterium *R. sulfidophilum* and demonstrated photoheterotrophic expression of artificial spider silk protein and silk fiber formation in this system and, more importantly, under photoautotrophic growth condition. Future work is needed to improve cell growth and protein expression under photoautotrophic growth conditions through methods such as supplementing seafood processing wastewater[53] into ASW medium and modifying the recombinant protein expression system. In principle, this marine photosynthetic microbial cell factory should also be suitable for the production of other biocompounds, which will contribute greatly to research communities and society in efforts to promote green, sustainable, and cost-effective bioprocesses.

## Methods

**Bacterial strains and cultivation conditions**. The marine photosynthetic purple nonsulfur bacterium *R. sulfidophilum* DSM1374/ATCC35886/W4[54,55] was obtained from the American Type Culture Collection (ATCC). For general cultivation purposes, *R. sulfidophilum* was maintained under photoheterotrophic growth conditions on marine agar (MA) or in MB (BD Difco) at 30 °C with continuous far-red LED light (730 nm, 20–30 W m$^{-2}$). Culture medium for recombinant strains of *R. sulfidophilum* was supplemented with 100 mg L$^{-1}$ kanamycin for plasmid maintenance purposes. *E. coli* DH5α (TaKaRa Bio) was used for general cloning purposes and maintained on lysogeny broth (LB) agar or in LB (BD Difco) at 37 °C under aerobic conditions with shaking at 180 rpm. For the purpose of plasmid conjugation into *R. sulfidophilum*, *E. coli* S17-1[56] was used as a donor strain and maintained in the same way as *E. coli* DH5α.

**Plasmid construction and conjugation into *R. sulfidophilum***. PCR amplifications of the *trc* promoter and *MaSp1* gene with various sizes of repeats (1-mer, 2-mer, 3-mer, and 6-mer) were performed using KOD-plus DNA polymerase (TOYOBO) with primers that added suitable restriction sites (shown in Supplementary Table 1). The *trc* promoter was amplified from a plasmid provided by Arikawa and Matsumoto (2016)[57]. The RBS sequence "AGGAGA", derived from the region upstream of the *pufQ* gene or *puf* operon (which encodes a photosynthetic

apparatus in *R. sulfidophilum*[38]), was added downstream of the *trc* promoter. *E. coli*-codon-optimized *MaSp1* gene sequences from the spider *N. clavipes*, together with His-Tag, S-Tag, thrombin and enterokinase sequences, were amplified from pET30-a-MaSp1[24,58] (Supplementary Table 2). Both *trc* promoter and *MaSp1* gene sequences were digested with appropriate restriction enzymes, purified and then ligated to the broad-host-range vector pBBR1MCS-2[35] (Fig. 1a) with Ligation high Ver.2 (TOYOBO). Bacterial transformation of the newly constructed pBBR1-P$_{trc}$-MaSp1 into *E. coli* DH5α or *E. coli* S17-1 was performed according to standard protocols[59]. Positive transformants were detected by colony PCR and double-confirmed with DNA sequencing. Plasmid isolation was carried out using a QIAprep Spin Miniprep Kit (QIAGEN). Bacterial conjugation between *E. coli* S17-1 (donor) harboring recombinant plasmid and *R. sulfidophilum* (recipient) was performed as described by[60]. In brief, *E. coli* S17-1 harboring plasmid was inoculated into 5 mL of LB medium supplemented with 50 mg L$^{-1}$ kanamycin and incubated at 37 °C for 16 h at 180 rpm. *R. sulfidophilum* was inoculated into 15 mL of MB and incubated at 30 °C with continuous far-red light (730 nm, 30 W m$^{-2}$) for 30 h. Both bacterial cultures were centrifuged at 10,000 g for 3 min and resuspended in fresh culture medium (LB for *E. coli* S17-1 and MB for *R. sulfidophilum*). Then, a bacterial suspension mixture was prepared according to a 1:1 ratio of *E. coli* S17-1 and *R. sulfidophilum*. Approximately 200 μL of the cell mixtures was spotted on an MA plate and incubated at 30 °C with continuous far-red light (730 nm, 30 W m$^{-2}$) for 24 h. Then, the regrown cells were scraped out and resuspended with 5 mL of fresh MB. Approximately 100 μL of cell suspension was spread on MA containing 100 mg L$^{-1}$ kanamycin and 100 mg L$^{-1}$ potassium tellurite. The plate was incubated at 30 °C with continuous far-red light (730 nm, 30 W m$^{-2}$) for 7 days. Positive conjugants were detected by colony PCR and double-confirmed by DNA sequencing. Sequence analyses were performed using ApE (A plasmid Editor) version 2.0.50b3 and SnapGene Viewer version 4.3.10.

**Expression of MaSp1 under photoheterotrophic conditions.** Recombinant *R. sulfidophilum* containing pBBR1-P$_{trc}$-MaSp1 was precultured in 15 mL of MB supplemented with 100 mg L$^{-1}$ kanamycin for two days at 30 °C with continuous far-red light (730 nm, 30 W m$^{-2}$) until the OD$_{660}$ reached ~1.2. After that, ~5 mL (10% v/v) of inoculum was transferred into 45 mL of fresh MB supplemented with 100 mg L$^{-1}$ kanamycin for four days of incubation at 30 °C with continuous far-red light (730 nm, 30 W m$^{-2}$) until the OD$_{660}$ indicated that the stationary growth phase had been reached (OD$_{660}$ ~2.0). The bacterial cells were harvested by centrifugation at 10,000 g for 10 min at 4 °C, and the supernatant was discarded. For resuspension, 5 mL of lysis buffer (10 mM Tris, 8 M urea, and 100 mM NaH$_2$PO$_4$, pH 7.4) was added to the cell pellet for every 1 g of wet cells. The cell suspensions were stirred vigorously for 12 h and then centrifuged at 10,000 g for 30 min. Finally, total soluble proteins in the supernatant fraction were collected and then quantified by a Pierce™ BCA Protein Assay Kit (Thermo Fisher Scientific). Soluble proteins were resolved via SDS-PAGE by using a 16.5% Mini-PROTEAN® Tris-Tricine precast gel (Bio-Rad). The gel was first stained with fixation buffer [25% (v/v) ethanol and 15% (v/v) formaldehyde] for 30 min before proceeding with Coomassie Brilliant Blue (CBB)-R250 (FUJIFILM Wako) staining for 1 h.

Western blotting was performed by electrophoretically transferring proteins from an SDS-PAGE gel to an Immuno-Blot® PVDF (polyvinylidene difluoride) membrane (0.2 μm pore size) (Bio-Rad) using a Trans-Blot® SD Semi-Dry Transfer Cell (Bio-Rad). The blotting procedures were conducted according to the His•Tag® western reagents protocol (Novagen Biosciences). In brief, membrane staining with Ponceau S (Beacle) was performed after electroblotting to confirm successful protein transfer before proceeding with blocking with 5% (w/v) milk/PBS for 12 h. The membrane was first probed with 0.2 μg mL$^{-1}$ monoclonal His•Tag primary antibody (Merck Millipore) for 2 h and then probed with 0.1 μg mL$^{-1}$ goat anti-mouse IgG HRP secondary antibody (Abcam) for 2 h. Finally, the membrane was treated with SuperSignal™ West Pico PLUS Chemiluminescent Substrate (Thermo Fisher Scientific), and chemiluminescence images were taken using a LAS-3000 imager (Fujifilm). Band intensities on the western blots were analyzed using Fiji/ImageJ version 1.52p[61].

**Identification of the MaSp1 proteins.** After CBB staining, target bands at the appropriate molecular weight positions on SDS-PAGE gels were excised and digested with trypsin. The resulting peptides were measured using LC–MS/MS using a Q Exactive mass spectrometer (Thermo Fisher Scientific) at the RIKEN CSRS Biomolecular Characterization Unit. The LC–MS/MS data were searched against in-house protein databases using the MASCOT program (Matrix Science)[62]. The mass spectrometry proteomics data have been deposited to the ProteomeXchange Consortium (http://proteomecentral.proteomexchange.org) via the jPOST partner repository[63] with the dataset identifier PXD019821.

**Expression of MaSp1 under photoautotrophic growth conditions.** Preculture was prepared in 15 mL of MB supplemented with 100 mg L$^{-1}$ kanamycin for 2 days at 30 °C with continuous far-red light (730 nm, 30 W m$^{-2}$) until the OD$_{660}$ reached ~1.2. After that, the cells were harvested by centrifugation at 10,000 g for 10 min, washed twice with 2% sodium chloride and resuspended in 15 mL of 2% sodium chloride before being transferred to new medium with ~2 mL (10% v/v) of inoculum. For photoautotrophic growth conditions, *R. sulfidophilum* was cultured

in 20 mL of Daigo's ASW SP medium for marine microalgae medium (Fujifilm) supplemented with 100 mg L$^{-1}$ kanamycin in a 20 mL tempered hard-glass gas chromatography vial with a rubber stopper cap (Nichiden-Rika Glass) at 30 °C with continuous far-red LED light (730 nm, 20–30 W m$^{-2}$) for 7 days. One liter of Daigo's ASW SP medium contains 9474 mg MgCl$_2$·6H$_2$O, 1326 mg CaCl$_2$·2H$_2$O, 3505 mg Na$_2$SO$_4$, 597 mg KCl, 171 mg NaHCO$_3$, 85 mg KBr, 34 mg Na$_2$B$_4$O7·10H$_2$O, 12 mg SrCl$_2$, 3 mg NaF, 1 mg LiCl, 0.07 mg KI, 0.0002 mg CoCl$_2$·6H$_2$O, 0.008 mg AlCl$_3$·6H$_2$O, 0.005 mg FeCl$_3$·6H$_2$O, 0.0002 mg Na$_2$WO$_4$·2H$_2$O, 0.02 mg (NH$_4$)$_6$Mo7O$_{24}$·4H$_2$O, 0.0008 mg MnCl$_2$·4H$_2$O and 20,747 mg NaCl, with the pH adjusted to 7.0 before autoclaving. Nitrogen was supplied daily by nitrogen gas bubbling at 0.5 L d$^{-1}$. The inorganic carbon was supplied as 1 g L$^{-1}$ sodium bicarbonate (NaHCO$_3$). Additional macronutrients, including 0.4 g L$^{-1}$ yeast extract, 2 mg L$^{-1}$ vitamin B12, 5 mg L$^{-1}$ ferric citrate and 0.5 g L$^{-1}$ KH$_2$PO$_4$, were added when necessary according to the experimental design. Cells were harvested by centrifugation at 10,000 g for 10 min, supernatant was discarded, and the cells were kept frozen at −80 °C before lyophilization with an FDU-2100 freeze dryer (EYELA) for 24 h. Finally, the CDM was measured.

**Large-scale production and purification of MaSp1.** Mass production of MaSp1-(6-mer) was performed with nine liters of MB supplemented with 100 mg L$^{-1}$ kanamycin in a 10 L jar fermenter (BEM) at 30 °C with continuous far-red LED light (850 nm, 15 W m$^{-2}$) at a stirrer speed of 50 rpm for 7 days, until the OD$_{660}$ reached ~2.0. Bacterial cells were harvested by centrifugation at 10,000 g for 10 min, and the supernatant was discarded. Soluble proteins were purified using a HisPrep™ FF 16/10 20 mL column (GE Healthcare Life Sciences) according to the manufacturer's protocol. Binding buffer (8 M urea, 0.5 M NaCl, 20 mM phosphate buffer, 40 mM imidazole, pH 7.4) and elution buffer (8 M urea, 0.5 M NaCl, 20 mM phosphate buffer, 500 mM imidazole, pH 7.4) were filtered through a 0.22 μm cellulose acetate filter (Corning) before use. After that, the eluent was concentrated using Amicon Ultra-15 6 MWCO 3000 Da centrifugal filters (Merck Millipore). The concentrated eluent was further purified with gel filtration chromatography via an ÄKTAexplorer (GE Healthcare Life Sciences) equipped with a Superdex®200 10/300 GL column, and the elution buffer contained 20 mM phosphate and 150 mM NaCl at pH 7.0. Fractions that contained the appropriate size of MaSp1-(6-mer) protein (kDa) were concentrated once again using Amicon Ultra-15 6 MWCO 3000 Da centrifugal filters before desalting with Milli-Q water using a PD-10 desalting column (GE Healthcare Life Sciences). The protein solution was frozen at −80 °C and then lyophilized for 24 h.

**Spider silk fiber extrusion and electron microscopy imaging.** Concentrated protein doping solution, 10 wt% MaSp1-(6-mer), was prepared by dissolving the lyophilized protein in 1,1,1,3,3,3-hexafluoro-2-propanol (HFIP) (Fujifilm Wako)[42]. The doping solution was extruded from a 100 μL QSP® gel loading pipette tip (inner orifice diameter ~0.13 mm) (Thermo Fisher Scientific) into a coagulation bath containing 90% (v/v) 2-propanol (Fujifilm Wako) on a glass petri dish, and it was pulled with an extra fine tip tweezer. The fibers were then dried at room temperature and further examined using a JCM-6000 Versatile Benchtop Scanning Electron Microscope (SEM) (JEOL).

**Statistics and reproducibility.** Data for MaSp1 expression in recombinant *R. sulfidophilum* including cell dry mass, % of MaSp1 in total proteins and MaSp1 yield were presented as the mean value ± standard deviation (SD). For comparisons among two or more groups, statistical significance was determined using a one-way analysis of variance (ANOVA), followed by Tukey's HSD post hoc tests using a statistical significance level of $P < 0.05$. Exact *p* values are available in[64]. All statistical analyses were carried out by Statistical Package for the Social Sciences (SPSS) software version 22 (IBM Corp. Released 22.0.0.0). Reproducibility of MaSp1 expression was evaluated using three independent biological replicates ($n = 3$).

## Data availability

All data that support the findings of this study are available as Supplementary Information files. LC–MS/MS analyses are available as Supplementary Data 1[39] (Figshare, https://doi.org/10.6084/m9.figshare.12502313). The mass spectrometry proteomics data are deposited to the ProteomeXchange Consortium (http://proteomecentral.proteomexchange.org) via the jPOST partner repository[63] with the dataset identifier PXD019821. The source data underlying the charts, plasmid DNA sequences, gel and blot images are available in Figshare[64–66] (https://doi.org/10.6084/m9.figshare.12495473; https://doi.org/10.6084/m9.figshare.12494507; https://doi.org/10.6084/m9.figshare.12495611).

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

## Acknowledgements

This study was funded by the ImPACT Program of the Council for Science, Technology and Innovation (Cabinet office, Government of Japan) and JST ERATO (Grant Number JPMJER1602). We would like to thank Dr. Takehiro Suzuki (RIKEN Center for Sustainable Resource Science, Biomolecular Characterization Unit) for his assistance in LC–MS/MS data acquisition.

## Author contributions

K.N. designed the research. C.P.F. and M.H.-T. performed photosynthetic bacterial experiments. A.D.M. performed the spinning experiments and the fiber evaluations. C.P.F. and N.A.O. prepared the plasmid constructs. C.P.F. and C.T. performed western blotting. C.P.F. and K.N. analyzed all the data. All the authors prepared the manuscript.

## Competing interests

The authors declare no competing interests.
