## [Peer Review File · Communications Biology]

Reviewers' comments:

Reviewer #1 (Remarks to the Author):

The manuscript by Numata and coworkers describes heterologous expression of a spider silk protein, spidroin MaSp, in a marine photosynthetic bacterium. The authors insist that they achieved protein expression in the photoautotrophic growth mode. However, notable expression was detected only when yeast extract was added to the medium. Hence, unfortunately, spidroin expression was achieved under photoheterotrophic growth conditions, not photoautotrophic conditions.

While there is a lot to like about this study, the goals for spidroin production in a phototrophic bacterium are somewhat unclear (reducing costs? increasing yield? What are the problems when using *E. coli*?). In this study, the authors compare medium ingredients, but there are more parameters to be tested especially when using phototrophs, e.g., light intensity, light wavelength, etc.

Description on the goals for the expression of repeat domains is insufficient. They tested monomer to 6-mer of the repeat domain of MaSp. Then, is 6-mer enough for their purpose? Is more multi-mer better? Although the 3-mer construct showed the highest expression in this study, the authors use the 6-mer for following experiments. Is the 3-mer not enough for what reasons?

Minor points:

Error bars are not provided in Fig. S1.

What samples were loaded for the western blot analysis shown in Fig. 2C?

Reviewer #2 (Remarks to the Author):

The article entitled "A Marine Photosynthetic Microbial Cell Factory as a Platform for Spider Silk Production" is well written and unique. The production of spider silk in photosynthetic bacterium is a concept that has been discussed in the spider silk community for a few years but this is the first report that this reviewer is aware of. The authors are correct in stating that the most remarkable claim of their study is that they have demonstrated a next-generation microbial cell factory that is based upon a marine photosynthetic organism utilizing photoautotrophic growth mode and renewable nonfood feedstocks and seawater. As proof of this statement, the authors successfully designed, cloned and then produced a synthetic spider silk protein, MaSp1, in this system and studied various parameters of growth and the subsequent result on protein production. Spider silks are notorious for being difficult to produce synthetically, as the authors note. Their successful production of the protein is significant and lends considerable credence that this system will be capable of producing other proteins. The authors do not overstate their data as it clearly indicates relatively low expression levels. However, expression did occur of this difficult protein and their work represents the first report in this system. Many years and considerable effort have gone into engineering *E. coli* and expression plasmids to get levels of spider silk protein produced to be in the 1g/l range as the authors acknowledge by way of reference. It is significant that they were able to achieve this level of production (3.93 ± 2.76 mg/l) in a novel system and that they were able to recover enough protein from which to spin fibers. The error bars on reported production levels are significant for most of the data but this reviewer believes this is acceptable if not expected for the first reported production of these difficult proteins in a new system. Further, western blots confirm the general trends of expression of the various clones in the different media conditions. The statistical analysis adds further credence to the data as being significantly different as indicated in the figures.

The conclusions are original and have not been reported to this reviewer's knowledge. Conclusions are well supported and convincing. As mentioned briefly above, this manuscript provides missing and important insight into the production of spider silk proteins in photosynthetic bacteria. Statistical analysis of their data is sufficient and the level of detail in material and methods is

adequate to reproduce the work.

The authors should be congratulated for their work on this novel system of producing recombinant spider silk and producing a well-written manuscript with carefully designed experiments and convincing data.

Date: April 3, 2020

Subject: Revised manuscript submission [Manuscript ID: COMMSBIO-20-0184-T]

To: Prof. Dr. Caitlin Karniski, Associate Editor, Communications Biology

From: Keiji Numata, RIKEN

Dear Dr. Caitlin Karniski,

We would like to submit revisions to our contribution entitled “A Marine Photosynthetic Microbial Cell Factory as a Platform for Spider Silk Production”. We considered all of each reviewer’s comments point-by-point and revised the manuscript based on the reviewers’ comments.

Reviewer #2 seems very positive about our results and outcomes reported here. Regarding the concerns from Reviewer #1, we have (i) revised the main texts for the sections related to spidroin expression in photoheterotrophic condition, (ii) described the advantages of our phototroph system compared to heterotrophs, and also (iii) described the goal for the expression of MaSp1 repeat domains. In addition, we also revised Fig. 2, where we included data points on the bar graphs to show data distribution. Sample size information was added to Fig. 2 and Supplementary Fig. 3 legends. A Data Availability statement is also added in the main text.

The revised parts are highlighted with yellow in the supporting information for review.

Thank you for your time and effort on behalf of our contribution.

Please address all correspondence to my address below.

With best regards,

Keiji Numata

Keiji Numata (corresponding author)

Biomacromolecules Research Team, RIKEN

2-1 Hirosawa, Wako-shi, Saitama 351-0198, Japan

Phone: +81-48-467-9525

Fax: +81-48-462-4664

Email: keiji.numata@riken.jp

Updated Fig. 2

Data points are included on the bar graphs to show data distribution. Sample size information was added to Fig. 2 legend.

Fig. 2: Photoautotrophic growth and heterotrophic production of artificial spider silk protein in the recombinant marine photosynthetic bacterium *Rhodovulum sulfidophilum*. **a**, Recombinant *R. sulfidophilum* harboring pBBR1-Ptrc-MaSp1-(6-mer) was cultivated using 20 mL of Daigo's artificial seawater (ASW) SP for marine microalgae medium in a 20 mL glass vial with a rubber stopper at 30°C with continuous far-red LED light (730 nm, 20 to 30 W m⁻²) for 7 days. Inorganic carbon was supplied as 1 g L⁻¹ sodium bicarbonate, while nitrogen was supplied via daily nitrogen gas bubbling at 0.5 L d⁻¹. Both marine broth (MB) and ASW media contained 100 mg L⁻¹ kanamycin. **b**, Biomass accumulation of recombinant *R. sulfidophilum* in various medium compositions based on cell dry mass (CDM). **c**, Semiquantitative quantification of MaSp1-(6-mer) expression in crude cell lysate of *R. sulfidophilum* by western blot using a monoclonal anti-His•Tag antibody. **d**, MaSp1-(6-mer) yield by recombinant *R. sulfidophilum* in various medium compositions. Mean data accompanied by different letters are significantly different with p-values < 0.05 (n = 3 biological replicates). (Abbreviation: C = NaHCO₃, YE = 0.4 g L⁻¹ yeast extract, N₂ = nitrogen gas and P = 0.5 g L⁻¹ KH₂PO₄).

Reviewer #1:

Comment 1: The manuscript by Numata and coworkers describes heterologous expression of a spider silk protein, spidroin MaSp, in a marine photosynthetic bacterium. The authors insist that they achieved protein expression in the photoautotrophic growth mode. However, notable expression was detected only when yeast extract was added to the medium. Hence, unfortunately, spidroin expression was achieved under photoheterotrophic growth conditions, not photoautotrophic conditions.

Response: Yes, we agreed partially with this point. We should correct the sentences and make it more precisely regarding the protein expression under photoautotrophic growth mode.

- In terms of photoautotrophic growth, the recombinant *Rhodovulum sulfidophilum* was capable to do so, as shown in the result of biomass accumulation in the culture with addition of N₂ gas, 1.0 g L⁻¹ NaHCO₃ and 0.5 g L⁻¹ KH₂PO₄ into artificial seawater (as shown in Fig. 2b below), which was significantly higher than culture with addition of 0.4 g L⁻¹ yeast.

Fig. 2: Photoautotrophic growth and heterotrophic production of artificial spider silk protein in the recombinant marine photosynthetic bacterium *Rhodovulum sulfidophilum*. **b**, Biomass accumulation of recombinant *R. sulfidophilum* in various medium compositions based on cell dry mass (CDM). Mean data accompanied by different letters are significantly different with p-values < 0.05 (n = 3 biological replicates). (Abbreviation: C = NaHCO₃, YE = 0.4 g L⁻¹ yeast extract, N₂ = nitrogen gas and P = 0.5 g L⁻¹ KH₂PO₄).

- In term of MaSp1 protein expression under photoautotrophic growth, there was no notable increment of MaSp1 expression (considering carry over of MaSp1 from inoculum). However, in the presence of 0.4 g L⁻¹ yeast extract (the only organic substance in the medium), notable increment of MaSp1 were observed in cultures (as shown in Fig. 2c below). As pointed out by the reviewer, suggested it as photoheterotrophic protein expression, we would like to define it as photoautotrophic growth with heterotrophic protein expression. The growth of *R. sulfidophilum* was mainly contributed by phosphorus (as mentioned above), while protein synthesis was mainly contributed by yeast extract.

Fig. 2: Photoautotrophic growth and heterotrophic production of artificial spider silk protein in the recombinant marine photosynthetic bacterium *Rhodovulum sulfidophilum*. **c**, Semiquantitative quantification of MaSp1-(6-mer) expression in crude cell lysate of *R. sulfidophilum* by western blot using a monoclonal anti-His•Tag antibody. Mean data accompanied by different letters are significantly different with p-values < 0.05 (n = 3 biological replicates). (Abbreviation: C = NaHCO₃, YE = 0.4 g L⁻¹ yeast extract, N₂ = nitrogen gas and P = 0.5 g L⁻¹ KH₂PO₄).

Along with the comment, we revised the text and figure legend.

- (page 2, line 13)

“Here, we have demonstrated the first heterotrophic production of spider dragline silk proteins in a marine photosynthetic purple bacterium, *Rhodovulum sulfidophilum*, under both photoheterotrophic and photoautotrophic growth conditions.”

- (page 4, line 65)

“... is capable of producing the hydrophobic repetitive sequence of MaSp1 using small amount of organic substance under photoheterotrophic or photoautotrophic growth conditions.”

- Fig. 2 (page 8, line 184)

“Photoautotrophic growth and heterotrophic production of artificial spider silk protein ...”

- (page 10, line 236)

“...we demonstrated photoautotrophic growth and heterotrophic synthesis of silk proteins by using mainly renewable nonfood feedstocks, small amount of organic substance and seawater as the cultivation medium.”

- (page 12, line 283)

“... demonstrated the first photoheterotrophic expression of artificial spider silk protein and silk fiber formation in this system and, more importantly, under photoautotrophic growth condition.”

- (page 16, line 377)

Photoautotrophic growth and heterotrophic production of spider dragline silk protein

- (Supplementary Fig.3)

Semiquantitative measurement of MaSp1-(6-mer) expression in recombinant *R. sulfidophilum* under photoheterotrophic and photoautotrophic growth conditions (as in Fig. 2c-d).

Comment 2: While there is a lot to like about this study, the goals for spidroin production in a phototrophic bacterium are somewhat unclear (reducing costs? increasing yield? What are the problems when using *E. coli*?).

Response: One of the main goals are to reduce the production cost of spider silk with sustainable bioproduction system. Raw materials that used in microbial fermentation could contribute up to 70% of production cost (Singh et al. 2017). Heterotrophs such as *E. coli* and *P. pastoris* require rich media that contain suitable organic carbon and nitrogen sources to support their growths. The use of phototrophs such as marine photosynthetic purple bacteria that only require inorganic minimal media for growth could reduce raw materials' cost. Besides, marine photosynthetic purple bacteria are capable of growing in seawater, which eliminates the use of freshwater, and thus could also reduce raw materials' cost.

- Along with the comment, we revised the text. (page 3, line 60).

“Besides, high price of spider silk due to high production cost also remains a challenge to be resolved. Raw materials that used in heterotrophic microbial fermentation systems could contribute up to 70% of production cost³². Here, we have developed an economical and sustainable marine photosynthetic microbial cell factory using *R. sulfidophilum*, ...”

Reference added:

Singh, V., Haque, S., Niwas, R., Srivastava, A., Pasupuleti, M. & Tripathi, C.K.M. Strategies for fermentation medium optimization: an in-depth review. *Front. Microbiol.* **7**, 2087 (2017).

Comment 3: In this study, the authors compare medium ingredients, but there are more parameters to be tested especially when using phototrophs, e.g., light intensity, light wavelength, etc.

Response: Yes, we agreed with the reviewer's comment, there are more parameters need to be tested for phototrophs. In our previous study, we had tried to examine the cell growth and polyhydroxyalkanoate (PHA) production in 520 medium under different light conditions, such as intensity (8 W/m² and 50 W/m²) and wavelengths (730nm, 800 nm and 850nm) (Higuchi and Numata, 2019).

As shown in figures below, the unpublished data belongs to the same experiment conducted in Fig B. The result showed that the cell dry masses were comparable for cultures under light conditions 800 nm & 50 Wm⁻², 730 nm & 50 Wm⁻², and 800 nm & 8 Wm⁻² from Day-3 to Day-

4 (unpublished data). Meanwhile, light conditions 800 nm & 8 Wm⁻² contributed the highest PHA concentration from Day-2 to Day-4 (Fig. B), followed by 730 nm & 8 Wm⁻² (Day-4) and 730 nm & 50 Wm⁻² (Day-4).

In this study, we used light conditions 730 nm & 20-30 Wm⁻², which are suitable for optimum cell growth. However, MaSp1 protein production and PHA production involved different metabolic pathways. Further optimization of light wavelength, intensity and cultivation time is required for MaSp1 protein production in this new *R. sulfidophilum* phototrophic expression host.

Reference Fig (Higuchi and Numata, 2019). Cell growth and PHA production under different light conditions. (B) PHA concentrations (mg/L) under different light conditions. Cells were cultured under low-light (LL: 8 W/m²), intermediate-light (ML: 20 W/m², 850 nm) and high-light (HL: 50 W/m²) conditions. PHA concentrations were determined after 1 to 4 days of incubation. Data are the mean ± SD of at least three cultures. **(unpublished data)** Cell dry mass (g/L) under different light conditions.

- Along with the comment, we revised the text. (page 9, line 200).

“In our previous study, we had examined the cell growth of *R. sulfidophilum* under different light conditions, such as intensity (8 W m⁻² and 50 W m⁻²) and wavelength (730 nm, 800 nm and 850 nm)⁴⁴. In this study, we evaluated ...”

Reference added:

Higuchi-Takeuchi, M. & Numata K. Acetate-inducing metabolic states enhance polyhydroxyalkanoate production in marine purple non-sulfur bacteria under aerobic conditions. *Front. Bioeng. Biotechnol.* **7**, 118 (2019).

Comment 4: Description on the goals for the expression of repeat domains is insufficient. They tested monomer to 6-mer of the repeat domain of MaSp. Then, is 6-mer enough for their purpose? Is more multi-mer better?

Response: The ultimate goal is to express MaSp1 protein with repeat/size similar to native dragline spider silk, which is >100-mer or >300 kDa. In general, the size of the spidroins have positive correlation to tensile strength until a certain size of molecular weight. Large proteins possess more interchain and intrachain interactions, entanglements, and less chain-end defects (Nunes et al. 1982; Xia et al. 2010; Bowen et al. 2018). The purpose of this study is to develop a new marine phototroph for spider silk production. Since we are using *R. sulfidophilum* phototrophic expression host for the first time, it would be great to start with small molecular weight of MaSp1. As discussed in previous literatures, there are many challenges need to be addressed when we perform heterologous expression of large spidroins such as genetic instability of long and highly repetitive DNA sequences, translation inhibition by complex mRNA secondary structures, a high demand for glycine and alanine tRNAs, and overall metabolic burden caused by spidroin overexpression (Xia et al. 2010; Chung et al. 2012; Bowen et al. 2018). Therefore, much a lot of works will be expected to optimize this new *R. sulfidophilum* phototrophic expression host for the production of much larger spidroins such as 300 kDa.

- To add these explanations, we revised the text. (page 7, line 147)

“Further attempts in expressing artificial spider silk proteins with sizes close to the native spider dragline silk (~100-mer or ~300 kDa), which had been achieved in metabolically engineered *E. coli* would be applicable in *R. sulfidophilum* as well. However, there are many challenges need to be resolved for the host in advance such as metabolic capability (high demand for glycine and alanine tRNAs) and stability of the genetic constructs (long and highly repetitive DNA sequences)^{41, 42}.”

- (page 10, line 242)

“In general, the size of the spidroins have positive correlation to tensile strength until a certain size of molecular weight. Large proteins possess more interchain and intrachain interactions, more entanglements, and less chain-end defects^{41, 42, 50}.”

References added:

Nunes, R.W., Martin, J.R. & Johnson, J.F. Influence of molecular weight and molecular weight distribution on mechanical properties of polymers. *Polym. Eng. Sci.* **22**, 205–228 (1982).

Chung, H., Kim, T.Y. & Lee, S.Y. Recent advances in production of recombinant spider silk proteins. *Curr. Opin. Biotechnol.* **23**, 957–964 (2012).

Bowen, C.H., Dai, B., Sargent, C.J., Bai, W., Ladiwala, P., Feng, H., Huang, W., Kaplan, D.L., Galazka, J.M. & Zhang, F. Recombinant spidroins fully replicate primary mechanical properties of natural spider silk. *Biomacromolecules* **19**, 3853-3860 (2018).

Comment 5: Although the 3-mer construct showed the highest expression in this study, the authors use the 6-mer for following experiments. Is the 3-mer not enough for what reasons?

Response: As we have mentioned in the comment #4, the higher the molecular weight of MaSp1 would contribute more tensile strength to the spider silk fiber. Therefore, we had decided to proceed with 6-mer for the subsequent experiments.

- To add this explanation, we revised the text. (page 7, line 160).

The largest repeat, MaSp1-(6-mer), was chosen for subsequent experiments because higher molecular weight of MaSp1 would contribute more tensile strength to the spider silk fiber.

Comment 6: Error bars are not provided in Fig. S1.

Response: This result is only for quick estimation of MaSp1 proteins (with different repeats, 1-mer, 2-mer, 3-mer and 6-mer) expression. We did not need to discuss this result quantitatively, just need to confirm the expression of MaSp1 proteins.

- Along with the comment, we revised the text. (page 6, line 138).

“In addition to the confirmation of MaSp1 proteins expression, we also performed a brief estimation of the amount of MaSp1 proteins obtained from the recombinant *R. sulfidophilum* cultures, which was approximately 3 to 10 mg L⁻¹ ...”

Comment 7: What samples were loaded for the western blot analysis shown in Fig. 2C?

Response: These are crude proteins or cell lysates from the recombinant *Rhodovulum sulfidophilum*, which containing our target protein MaSp1-(6-mer). Sample names loaded for the western blot analysis are shown in Fig. 2C (upper part), which are same sample names for the bar chart (bottom part).

Fig. 2: Photoautotrophic growth and heterotrophic production of artificial spider silk protein in the recombinant marine photosynthetic bacterium *Rhodovulum sulfidophilum*. c, Semiquantitative quantification of MaSp1-(6-mer) expression in crude cell lysate of *R. sulfidophilum* by western blot using a monoclonal anti-His-Tag antibody. Mean data accompanied by different letters are significantly different with p-values < 0.05 (n = 3 biological replicates). (Abbreviation: C = NaHCO₃, YE = 0.4 g L⁻¹ yeast extract, N₂ = nitrogen gas and P = 0.5 g L⁻¹ KH₂PO₄).

- Along with the comment, we revised the Fig. 2 legend. (page 8, line 193)
- c. Semiquantitative quantification of MaSp1-(6-mer) expression in crude cell lysate of *R. sulfidophilum* by western blot using a monoclonal anti-His•Tag antibody.

Reviewer #2:

Comment 1: The article entitled “A Marine Photosynthetic Microbial Cell Factory as a Platform for Spider Silk Production” is well written and unique. The production of spider silk in photosynthetic bacterium is a concept that has been discussed in the spider silk community for a few years but this is the first report that this reviewer is aware of.

The authors are correct in stating that the most remarkable claim of their study is that they have demonstrated a next-generation microbial cell factory that is based upon a marine photosynthetic organism utilizing photoautotrophic growth mode and renewable nonfood feedstocks and seawater. As proof of this statement, the authors successfully designed, cloned and then produced a synthetic spider silk protein, MaSp1, in this system and studied various parameters of growth and the subsequent result on protein production. Spider silks are notorious for being difficult to produce synthetically, as the authors note. Their successful production of the protein is significant and lends considerable credence that this system will be capable of producing other proteins.

Response: Thank you for the encouraging comments and compliments.

Comment 2: The authors do not overstate their data as it clearly indicates relatively low expression levels. However, expression did occur of this difficult protein and their work represents the first report in this system. Many years and considerable effort have gone into engineering *E. coli* and expression plasmids to get levels of spider silk protein produced to be in the 1g/l range as the authors acknowledge by way of reference. It is significant that they were able to achieve this level of production (3.93±2.76 mg/l) in a novel system and that they were able to recover enough protein from which to spin fibers.

Response: Thank you for the encouraging comments and compliments.

Comment 3: The error bars on reported production levels are significant for most of the data but this reviewer believes this is acceptable if not expected for the first reported production of these difficult proteins in a new system. Further, western blots confirm the general trends of expression of the various clones in the different media conditions. The statistical analysis adds further credence to the data as being significantly different as indicated in the figures.

Response: Thank you for the encouraging comments and compliments.

Comment 4: The conclusions are original and have not been reported to this reviewer's knowledge. Conclusions are well supported and convincing. As mentioned briefly above, this manuscript provides missing and important insight into the production of spider silk proteins in photosynthetic bacteria.

Response: Thank you for the encouraging comments and compliments.

Comment 5: Statistical analysis of their data is sufficient and the level of detail in material and methods is adequate to reproduce the work. The authors should be congratulated for their work on this novel system of producing recombinant spider silk and producing a well-written manuscript with carefully designed experiments and convincing data.

Response: Thank you for the encouraging comments and compliments.

REVIEWERS' COMMENTS:

In the original manuscript, the authors insisted that the spider silk protein was produced under photoautotrophic growth conditions, but I pointed out that it was not actually achieved. The authors seem to admit it in their "rebuttal" letter and changed the phrase photoautotrophic all to photoheterotrophic in the revised manuscript.

Because my decision was made mostly based on the point that the achievement under photoautotrophic conditions is crucial for the quality and significance of the paper, I cannot recommend the paper for publication in Communications Biology.

As to other concerns I made, for instance, the significance of the expression of the repeated domain, the author thoroughly corrected them and the manuscript becomes clearer.